# Adverse cardiovascular events and cardiac imaging findings in patients on immune checkpoint inhibitors

Jennifer M. Kwan[1,2☯], Miles Shen[1,2☯], Narjes Akhlaghi[2,3], Jiun-Ruey Hu[1,2], Ruben Mora[4], James L. Cross[2], Matthew Jiang[2,3], Michael Mankbadi[2,3], Peter Wang[2,3], Saif Zaman[2,3], Seohyuk Lee[2], Yunju Im[5,6], Attila Feher[1,2], Yi-Hwa Liu[2], Shuangge S. Ma[6], Weiwei Tao[6], Wei Wei[6], Lauren A. Baldassarre[1,2]*

1 Section of Cardiovascular Medicine, Yale University School of Medicine, New Haven, Connecticut, United States of America, 2 Yale University School of Medicine, New Haven, Connecticut, United States of America, 3 Department of Internal Medicine, Yale University School of Medicine, New Haven, Connecticut, United States of America, 4 Nuvance Health, Danbury, Connecticut, United States of America, 5 Department of Biostatistics, University of Nebraska Medical Center, Omaha, Nebraska, United States of America, 6 Department of Biostatistics, Yale University School of Public Health, New Haven, Connecticut, United States of America

☯ These authors contributed equally to this work.
* lauren.baldassarre@yale.edu

**Data Availability Statement:** All relevant data are within the manuscript and its Supporting Information files.

## Abstract

### Background

There is an urgent need to better understand the diverse presentations, risk factors, and outcomes of immune checkpoint inhibitor (ICI)-associated cardiovascular toxicity. There remains a lack of consensus surrounding cardiovascular screening, risk stratification, and clinical decision-making in patients receiving ICIs.

### Methods

We conducted a single center retrospective cohort study including 2165 cancer patients treated with ICIs between 2013 and 2020. The primary outcome was adverse cardiovascular events (ACE): a composite of myocardial infarction, coronary artery disease, stroke, peripheral vascular disease, arrhythmias, heart failure, valvular disease, pericardial disease, and myocarditis. Secondary outcomes included all-cause mortality and the individual components of ACE. We additionally conducted an imaging substudy examining imaging characteristics from echocardiography (echo) and cardiac magnetic resonance (CMR) imaging.

### Results

In our cohort, 44% (n = 962/2165) of patients experienced ACE. In a multivariable analysis, dual ICI therapy (hazard ratio [HR] 1.23, confidence interval [CI] 1.04–1.45), age (HR 1.01, CI 1.00–1.01), male sex (HR 1.18, CI 1.02–1.36), prior arrhythmia (HR 1.22, CI 1.03–1.43), lung cancer (HR 1.17, CI 1.01–1.37), and central nervous system (CNS) malignancy (HR 1.23, CI 1.02–1.47), were independently associated with increased ACE. ACE was

**Funding:** This work was supported by: 1) LB, grant #18CDA34110361, American Heart Association (https://www.heart.org). 2) JMK, CTSA grant #KL2 TR001862, National Center for Advancing Translational Science (NCATS) (https://ncats.nih. gov/), a component of the National Institutes of Health (NIH). Funders did not play any role in the study design, data collection and analysis, decision to publish, or preparation of the manuscript.

**Competing interests:** The authors have declared that no competing interests exist.

independently associated with a 2.7-fold increased risk of mortality ($P$<0.001). Dual ICI therapy was also associated with a 2.0-fold increased risk of myo/pericarditis ($P$ = 0.045), with myo/pericarditis being associated with a 2.9-fold increased risk of mortality ($P$<0.001). However, the cardiovascular risks of dual ICI therapy were offset by its mortality benefit, with dual ICI therapy being associated with a ~25% or 1.3-fold decrease in mortality. Of those with echo prior to ICI initiation, 26% (n = 115/442) had abnormal left ventricular ejection fraction or global longitudinal strain, and of those with echo after ICI initiation, 28% (n = 207/740) had abnormalities. Of those who had CMR imaging prior to ICI initiation, 43% (n = 9/21) already had left ventricular dysfunction, 50% (n = 10/20) had right ventricular dysfunction, 32% (n = 6/19) had left ventricular late gadolinium enhancement, and 9% (n = 1/11) had abnormal T2 imaging.

## Conclusion

Dual ICI therapy, prior arrhythmia, older age, lung and CNS malignancies were independently associated with an increased risk of ACE, and dual ICI therapy was also independently associated with an increased risk of myo/pericarditis, highlighting the utmost importance of cardiovascular risk factor optimization in this particularly high-risk population. Fortunately, the occurrence of myo/pericarditis was relatively uncommon, and the overall cardiovascular risks of dual ICI therapy appeared to be offset by a significant mortality benefit. The use of multimodal cardiac imaging can be helpful in stratifying risk and guiding preventative cardiovascular management in patients receiving ICIs.

## Introduction

Over the past decade, immune checkpoints inhibitors (ICIs) have revolutionized the landscape of cancer care. Currently, eleven ICIs are approved by the Food and Drug Administration for the treatment of various malignancies, including but not limited to melanoma, non-small cell lung cancer, renal cell carcinoma, urothelial cancer, head and neck squamous cell carcinoma, Hodgkin's lymphoma, gastrointestinal cancers, breast cancer, and cancers with high microsatellite instability or defective mismatch repair [1–3]. Combination ICI therapies are also used to increase anti-tumor activity [4].

Through the inhibition of cytotoxic T lymphocyte-associated protein-4 (CTLA-4), programmed cell death protein-1 (PD-1), and programmed cell death 1 ligand-1 (PD-L1), ICIs bolster the host's immune system to effectively recognize and target tumor cells [2]. While ICIs are effective for a growing number of cancer patients, the enhanced T-cell activity against host tissues can lead to a wide range of immune-related adverse events (irAEs). Adverse cardiovascular events (ACE) were initially underappreciated in prospective trials of ICIs, likely in part due to lack of standardized cardiac monitoring and testing, nonspecific clinical manifestations, and difficulties in diagnosis, compared to other irAEs such as pneumonitis, colitis, or hepatitis [5–11]. However, since the approval of ICIs, cardiovascular toxicities have been increasingly reported [8,12–16]. There is still much to learn about the unique presentations, risk factors, and outcomes of ICI-associated cardiovascular toxicity. The existing literature on ICI-associated cardiovascular toxicity has in large part focused on myocarditis, with these studies describing fulminant and fatal presentations; however, other studies have also reported on smoldering and asymptomatic cases [13,17–23]. Other less-commonly reported presentations of ICI-associated

cardiovascular toxicities include pericardial disease, vasculitis, takotsubo-like syndrome, heart failure, myocardial infarction, coronary vasospasm, and arrhythmias [16,23–31]. ICIs have also been implicated in the acceleration of atherogenesis and, importantly, have been associated with increased risk of major adverse cardiovascular events [32–37].

Among risk factors, combination ICI therapy is most strongly correlated with incidence and severity of ICI-associated myocarditis [9,13,22]. Other possible risk factors suggested by epidemiologic and cohort studies for ICI-associated cardiovascular toxicity include concomitant use of cardiotoxic anti-neoplastic agents such as anthracyclines, prior radiation therapy, underlying cardiovascular disease, underlying autoimmune disease, tumor-related factors, concurrent irAEs such as skeletal myositis, and genetic factors [20,21,38–42].

With expanding indications for ICI therapy, an increasing number of patients are eligible to receive these agents, including an increasing number of patients with preexisting cardiovascular risk factors [3,43]. As such, further efforts to improve our understanding of ICI-associated cardiotoxicity are needed in order to guide clinical decision-making and to improve patient outcomes.

Cardiac imaging modalities such as echocardiography (echo) and cardiac magnetic resonance (CMR) imaging are integral to the evaluation of cardiotoxicity from ICI. Both echo and CMR provide valuable assessment of cardiac structure and function. CMR additionally provides the added value of tissue characterization, where the presence of late gadolinium enhancement (LGE) can be indicative of myocardial injury or fibrosis, and increased T2 signal can be indicative of myocardial edema. Abnormalities in both T1- and T2-weighted imaging can be seen in both ischemic and nonischemic processes, including from prior or current cancer therapies, such as chemotherapy and radiation therapy [44]. CMR can also aid in the diagnosis of acute myocarditis with the application of the Lake Louise Criteria, which incorporates main criteria (abnormal T1 or T2 parameters), with supportive criteria (pericarditis or left ventricular dysfunction) [45]. We conducted a large cohort study evaluating the presentations, risk factors, ACE outcomes, in patients who received ICI therapy. We additionally performed an imaging substudy to compare imaging characteristics on echo and CMR pre- and post-ICI.

## Methods

### Study population and covariates of interest

We performed a retrospective cohort study including cancer patients treated with ICIs between 2013 to 2020 at a large academic institution (Yale New Haven Hospital, Connecticut). The study was approved by the Yale University Institutional Review Board (IRB #2000026073) and a waiver for informed consent was obtained. Data was collected from electronic health records by our institutional Joint Data Analytics Team on July 19, 2019 and was stored in a HIPAA-compliant data repository. No members of the Joint Data Analytics Team were involved in the study design, statistical analyses, and manuscript preparation. Authors had access to information that could identify individual participants during or after data collection. Covariates of interest included patient demographics, cardiovascular risk factors including hypertension (HTN), hyperlipidemia (HLD), diabetes mellitus (DM), smoking history, chronic kidney disease (CKD), and rheumatologic disorders. Cardiovascular conditions included atherosclerotic cardiovascular disease (ASCVD), heart failure (HF), valvular disease, arrhythmias, pericardial disease, endocarditis, and myocarditis. Data pertaining to cancer included cancer types, presence of metastatic disease, specific ICI agents, and dual ICI therapy, which was defined by concurrent use of two ICI agents. For the imaging substudy, echo and CMR data were extracted from the database associated with our local picture archiving and

communication system (PACS). Cardiac imaging was obtained when clinically indicated, and all pre-ICI scans were compared with post-ICI scans.

## Study outcomes

The primary outcome was the occurrence of ACE, defined as a composite of ASCVD—which included coronary artery disease (CAD), myocardial infarction (MI), stroke (CVA), and peripheral arterial disease (PAD)—, heart failure (HF), arrhythmia (includes supraventricular tachycardia, atrial fibrillation, atrial flutter, ventricular tachycardia, ventricular fibrillation)), valvular disease (moderate or severe), pericardial disease, and myocarditis after initiation of ICI therapy. The secondary outcomes included the occurrence of the individual components of ACE and all-cause mortality. Time to event was recorded from the date the subject was started on ICI (Jan 1, 2013) to the first occurrence of each outcome or to the study end date after most recent ICI administration. Investigators also manually adjudicated a subset of 200 randomly-chosen patients to evaluate the accuracy of electronic health record data acquisition methodology, which demonstrated good correlation with overall similar comorbidities (**S1 Table**); 1% of patients developed myo/pericarditis, which was not statistically significant compared to the entire cohort ($P = 0.745$). For the imaging substudy, abnormal echo criteria included global longitudinal strain (GLS) >-18% and left ventricular ejection fraction (LVEF) <55%. Abnormal CMR criteria included presence of LGE, abnormal T1 or T2 mapping, LVEF <57%, or right ventricular ejection fraction (RVEF) <52%.

## Statistical analysis

Continuous variables were presented as means with standard deviations or medians with inter-quartile ranges and were compared using Student's t-test. Categorical variables were presented as counts and percentages and were compared using Chi square or Fisher's exact tests. Composite ACE, as well as its individual components, were analyzed in a competing risks model using specific ICI agents, dual ICI therapy, and additional covariates, including age, sex, cancer types, cardiovascular diseases, and other medical comorbidities. All-cause mortality was assessed using a Cox regression model with time-dependent covariates using ACE, myo/pericarditis (myocarditis, pericarditis, or both), and additional covariates, including age, sex, cancer types, cardiovascular diseases, and other medical comorbidities. To investigate the effect of ICI on ACE compared to patients who did not receive ICI, we performed propensity score matching between the ICI cohort and a non-ICI cohort (**S1 Fig**). The non-ICI cohort was constructed from patients who received tyrosine kinase inhibitors (TKI) for three cancer types (lung, GI, and renal) that are commonly treated with ICI. We used optimal pair matching which forms matched pairs such that the average absolute distance across all the matched pairs is minimized. Cox regression was performed on the propensity-matched dataset using ICI status, age, sex, select cardiovascular diseases, and other medical comorbidities as covariates. All tests were considered statistically significant at $P<0.05$. All statistical analyses were performed using R software, version 4.1.0 (Vienna, Austria). Relevant imaging parameters from echo and CMR were extracted from imaging reports using natural language processing (NLP) techniques–the NLP pipeline was scripted using the NLTK toolkit in Python (v3.11.5) [46].

## Results

A total of 2165 patients who received any ICI for any type of cancer between 2013 to 2020 were included in this study. Median follow-up duration was 500 days (IQR 886 days). Of these patients, 962 (44%) developed ACE. Patients who developed ACE were older compared to those who did not develop ACE (mean age 70.4±11.6 vs 68.7±12.4, $P = 0.001$) and more

patients who developed ACE had a prior history of arrhythmia compared to those who did not develop ACE (27.3% vs 20.2%, $P<0.0001$). Patients who developed ACE also had a higher rate of prior HF compared to patients who did not develop ACE (12.4% vs 8.7%, $P = 0.007$). HTN (67.0% vs 61.4%, $P = 0.007$) and HLD (44.5% vs 38.8%, $P = 0.008$) were more prevalent in those who developed ACE compared to those who did not develop ACE (**Table 1**).

The most frequently used ICI, either as a single agent or combined with other ICIs, was pembrolizumab (n = 773), followed by nivolumab (n = 670). Furthermore, there was a higher incidence of ACE in the cohort of patients receiving dual ICI therapy compared to those who received single ICI therapy (50.9% vs 43.2%, $P = 0.011$). Out of all ICI agents, ipilimumab was associated with the highest rate of ACE (52.2%). The occurrences of ACE and its individual components by ICI agent are shown in **Table 2**.

**Table 1. Baseline demographics and comorbidities by presence of post-treatment adverse cardiovascular events.** Baseline demographics and comorbidities of patients who did not experience ACE compared to those who experienced ACE.

| | TOTAL (N = 2165) | NO ACE (N = 1203) | ACE (N = 962) | P-VALUE |
|---|---|---|---|---|
| **Demographic Characteristics** | | | | |
| Age - Mean ± SD (years) | 69.4 ± 12.1 | 68.7 ± 12.4 | 70.4 ± 11.6 | **0.001** |
| BMI - Mean ± SD (kg/m$^2$) | 27.0 ± 3.98 | 27.0 ± 3.98 | 27.1 ± 3.96 | 0.270 |
| Male - % (n/N) | 56.3% (1218/2165) | 53.0% (637/1203) | 60.4% (581/962) | **0.001** |
| Hispanic - % (n/N) | 3.9% (82/2121) | 3.5% (41/1168) | 4.3% (41/953) | 0.366 |
| Race - % (n/N) | | | | 0.400 |
| White | 86.6% (1875/2164) | 86.0% (1034/1202) | 87.4% (841/962) | |
| Black | 6.3% (136/2164) | 6.1% (73/1202) | 6.5% (63/962) | |
| Asian | 1.2% (27/2164) | 1.4% (17/1202) | 1.0% (10/962) | |
| Other | 5.8% (126/2164) | 6.5% (78/1202) | 5.0% (48/962) | |
| **Comorbidities - % (n/N)** | | | | |
| Arrhythmia | 23.4% (506/2165) | 20.2% (243/1203) | 27.3% (263/962) | **<0.001** |
| ASCVD | 39.1% (846/2165) | 37.7% (454/1203) | 40.7% (392/962) | 0.156 |
| Heart failure | 10.3% (224/2165) | 8.7% (105/1203) | 12.4% (119/962) | **0.007** |
| Valvular disease | 8.8% (190/2165) | 7.7% (93/1203) | 10.1% (97/962) | 0.056 |
| Pericardial disease | 1.3% (29/2165) | 1.1% (13/1203) | 1.7% (16/962) | 0.263 |
| Myocarditis | 0.0% (1/2165) | 0.0% (0/1203) | 0.1% (1/962) | 0.444 |
| Hypertension | 63.9% (1384/2165) | 61.4% (739/1203) | 67.0% (645/962) | **0.007** |
| Hyperlipidemia | 41.3% (895/2165) | 38.8% (467/1203) | 44.3% (428/962) | **0.008** |
| Diabetes mellitus | 21.3% (461/2165) | 20.9% (252/1203) | 21.7% (209/962) | 0.673 |
| Ever smoker | 72.3% (1559/2156) | 71.4% (856/1199) | 73.5% (703/957) | 0.309 |
| Chronic kidney disease | 12.6% (273/2165) | 11.7% (141/1203) | 13.7% (132/962) | 0.171 |
| Venous thromboembolism | 7.4% (160/2165) | 7.1% (85/1203) | 7.8% (78/962) | 0.563 |
| Endocarditis | 0.1% (2/2165) | 0.1% (1/1203) | 0.1% (1/962) | >0.999 |
| Pulmonary disease[I] | 42.2% (914/2165) | 41.7% (502/1203) | 42.8% (412/962) | 0.630 |
| Endocrine disease[II] | 7.0% (151/2165) | 6.7% (81/1203) | 7.3% (70/962) | 0.671 |
| Rheumatologic disease[III] | 3.2% (69/2165) | 3.0% (36/1203) | 3.4% (33/962) | 0.623 |

Normally distributed continuous variables are presented as mean ± standard deviation. ASCVD: Atherosclerotic cardiovascular disease, including acute coronary syndrome (ACS), those with history of myocardial infarction (MI), stable or unstable angina or coronary or other arterial revascularization, stroke, transient ischemic attack (TIA), or peripheral artery disease (PAD).

[I] Pulmonary diseases including chronic obstructive pulmonary disease, interstitial lung disease, asthma.

[II] Endocrine diseases including thyroid, adrenal, pituitary gland dysfunction.

[III] Rheumatologic diseases including systemic lupus erythematosus, rheumatoid arthritis, scleroderma, myositis, mixed connective tissue disease, Sjögren's syndrome.

**Table 2. Adverse cardiovascular events by immune checkpoint inhibitor.** Occurrence of ACE and individual components of ACE, by single ICI versus dual ICI therapy, as well as by specific ICI agents.

| | ACE | ARRHYTHMIA | ASCVD | HEART FAILURE | VALVULAR DISEASE | PERICARDIAL DISEASE | MYOCARDITIS |
|---|---|---|---|---|---|---|---|
| Single ICI therapy | 43.2% (788/1823) | 20.2% (368/1823) | 27.5% (502/1823) | 11.1% (202/1823) | 6.4% (117/1823) | 1.9% (34/1823) | 0.5% (10/1823) |
| Dual ICI therapy | 50.9% (174/342) | 22.2% (76/342) | 31% (106/342) | 13.2% (45/342) | 7% (24/342) | 1.8% (6/342) | 2% (7/342) |
| *P*-value | **0.011** | 0.382 | 0.191 | 0.267 | 0.635 | 1.000 | **0.011** |

**ICI Agent**

| | ACE | ARRHYTHMIA | ASCVD | HEART FAILURE | VALVULAR DISEASE | PERICARDIAL DISEASE | MYOCARDITIS |
|---|---|---|---|---|---|---|---|
| Nivolumab | 41.6% (279/670) | 19.1% (128/670) | 28.4% (190/670) | 10.6% (71/670) | 6.1% (41/670) | 1.2% (8/670) | 0.3% (2/670) |
| Pembrolizumab | 43.1% (333/773) | 19.9% (154/773) | 26.8% (207/773) | 11.4% (88/773) | 7.1% (55/773) | 1.9% (15/773) | 0.9% (7/773) |
| Ipilimumab | 52.2% (96/184) | 20.1% (37/184) | 37% (68/184) | 9.8% (18/184) | 9.2% (17/184) | 1.1% (2/184) | 0.5% (1/184) |
| Atezolizumab | 44.3% (94/212) | 22.6% (48/212) | 24.5% (52/212) | 11.3% (24/212) | 5.2% (11/212) | 1.9% (4/212) | 0.5% (1/212) |
| Durvalumab | 37.9% (36/95) | 18.9% (18/95) | 25.3% (24/95) | 8.4% (8/95) | 4.2% (4/95) | 2.1% (2/95) | 0.0% (0/97) |
| Avelumab | 45.5% (5/11) | 9.1% (1/11) | 18.2% (2/11) | 0% (0/11) | 9.1% (1/11) | 18.2% (2/11) | 0.0% (0/12) |
| Cemiplimab | 40% (2/5) | 20.0% (1/5) | 0.0% (0/5) | 40.0% (2/5) | 0.0% (0/5) | 20.0% (1/5) | 0.0% (0/5) |

ASCVD: Atherosclerotic cardiovascular disease, including acute coronary syndrome (ACS), those with history of myocardial infarction (MI), stable or unstable angina or coronary or other arterial revascularization, stroke, transient ischemic attack (TIA), or peripheral artery disease (PAD).

The occurrence of ACE by cancer type is summarized in **S2 Table**, with top cancers represented including lung, melanoma, liver and gastrointestinal cancers. Kaplan Meier survival curves and median times to ACE and the individual components of ACE are shown in **S2 Fig**.

We additionally assessed the independent risk of select demographic characteristics, cardiovascular risk factors, medical comorbidities, and dual ICI therapy on ACE as well as the individual components of ACE. Dual ICI therapy (HR 1.23, CI 1.04–1.45), age (HR 1.01, CI 1.00–1.01), male sex (HR 1.18, CI 1.02–1.36), arrhythmia (HR 1.22, CI 1.03–1.43), lung cancer (HR 1.17, CI, 1.00–1.37), and CNS malignancy (HR 1.23, CI 1.02–1.47) were independently associated with increased occurrence of ACE (**Fig 1A**). Age (HR 1.02, CI 1.01–1.03), male sex (HR 1.30, CI 1.08–1.55), HLD (HR 1.25, CI 1.05–1.48), and CNS malignancy (HR 1.79, CI 1.45–2.21), were independently associated with increased occurrence of ASCVD. Smoking history was strongly associated with an increased occurrence of ASCVD (HR 1.22, CI 1.00–1.48); however, this barely did not meet significance with a *P* = 0.052. In contrast, a prior history of ASCVD (HR 0.72, CI 0.60–0.87) was associated with a lower risk of new or worsening ASCVD following ICI therapy (**Fig 1B**). History of valvular disease (HR 1.70, CI 1.15–2.51) and HF (HR 1.63, CI 1.13–2.36) were independently associated with an increased risk of new or worsening HF with ICI therapy (**Fig 1C**). History of arrhythmia (HR 1.48, CI 1.16–1.89), lung cancer (HR 1.40, CI 1.09–1.80), and genitourinary cancers (HR 1.29, CI 1.04–1.61) were independently associated with increased risk of new or worsening arrythmia (**Fig 1D**). History of valvular disease (HR 1.70, CI 1.03–2.79) and hematologic malignancies (HR 1.78, CI 1.07–2.95) were independently associated with an increased risk of new or worsening valvular disease (**Fig 1E**). Lastly, dual ICI therapy (HR 1.96, CI 1.01–3.74), head and neck cancer (HR 2.20, CI 1.08–4.49), and lung cancer (HR 3.45, CI 1.49–7.98) were associated with an increased risk of myo/pericarditis, whereas younger age marginally reduced the risk of myo/pericarditis (HR 0.98, CI 0.95–1.00) (**Fig 1F**).

We also assessed the independent risk of individual ICI agents on ACE as well as individual components of ACE (**S3 Fig**). Although there were varying degrees of statistical significance in comparison to the multivariable model in Fig 1, the hazard ratios for covariates largely trended

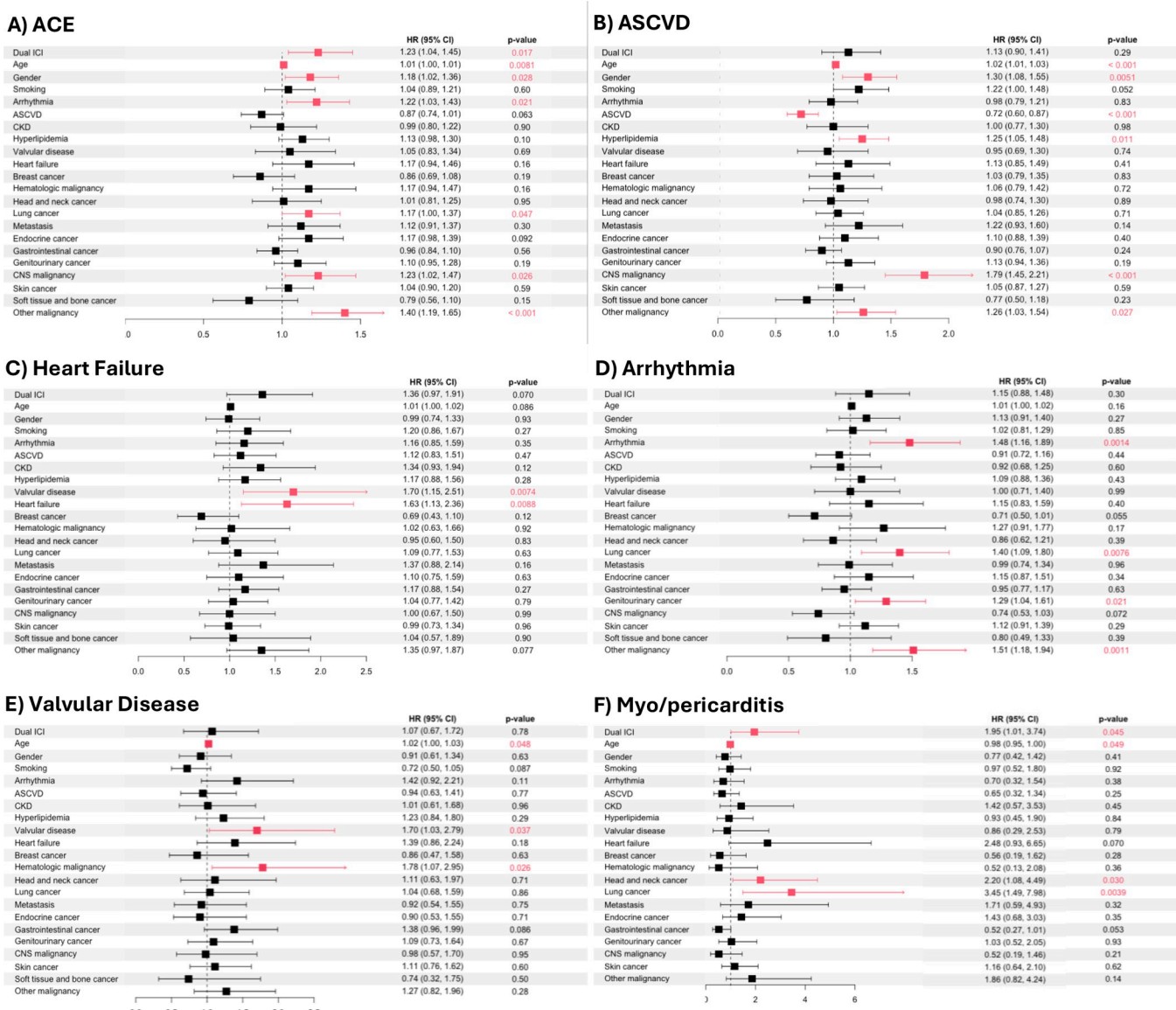

**Fig 1. Cox regression models for adverse cardiovascular events (ACE) and individual components of ACE.**

in the same direction. Although a high occurrence of ACE was previously observed in patients who received ipilimumab (**Table 2**), after adjusting for covariates in a multivariable Cox regression model, it was not independently associated with an increased risk of ACE, ASCVD, arrhythmia, or valvular disease (**S3 Fig**). Moreover, a propensity score matched analysis was performed to assess whether ICI use was independently associated with ACE in comparison to a matched cohort that did not receive ICI therapy (patients who received TKIs) (**S1 Fig**). There were a total of 1518 matched patients, with 759 patients in each group. The standardized mean differences for the matched variables were well below the 0.1 threshold, indicating that balance between the matched variables had been achieved (Figure A of **S1 Fig**). Based on the matched samples, ICI use was not an independent risk factor for ACE (Figure B of **S1 Fig**). Demographics and comorbidities of the propensity matched cohort are featured in **S3 Table**.

In a secondary analysis, we evaluated the effect of ACE and myo/pericarditis on all-cause mortality. In the extended Cox regression model with time varying covariates, ACE (**Fig 2A**)

independently increased the risk of mortality by 2.7-fold ($P<0.001$). Myo/pericarditis (**Fig 2B**) also independently increased risk of mortality by 2.9-fold ($P<0.001$). Pre-existing arrhythmia, metastatic disease, and ASCVD were also associated with increased risk of mortality (HR 1.18, $P = 0.024$; HR 1.29, $P = 0.032$; and HR 1.31, $P<0.001$, respectively). As previously noted, dual ICI therapy was associated with a higher risk of ACE and myo/pericarditis (**Fig 1A and 1F**). However, dual ICI therapy was also associated with a ~25% or 1.3-fold decrease in mortality ($P<0.001$) (**Fig 2**).

## Imaging substudy

Given the high prevalence of ACE in our cohort, we sought to evaluate echo and CMR imaging characteristics, with comparison of all pre- and post-ICI scans. Forty-four percent (n = 955/ 2165) of patients had an echo at some point in time. Forty-four percent of those patients (n = 442/955) had an echo prior to ICI initiation, of which 26% (n = 115/442) were abnormal. Median time from echo to time of first ICI administration was 112 days. Median LVEF was 50% (IQR 9.9%) and GLS was -15% (IQR 3.0%) prior to ICI initiation. Seventy-seven percent of patients (n = 740/955) had an echo after ICI initiation, of which 28% (n = 207/740) were abnormal (**S4 Fig**). Median time from last ICI administration to echo was 241 days. Median LVEF and GLS were 48% (IQR 12.2%) and -16% (IQR 3.0%) after ICI initiation, respectively. Of those who had abnormal echo's prior to ICI, 9.6% (n = 11/115) had an interval decline in LVEF after initiation of ICI. The reduction in LVEF trended toward significance ($P = 0.09$), but the change in GLS was not significantly different ($P = 0.57$) in comparing pre- and post-ICI scans.

There were 104 unique patients of the entire cohort of 2165 (5%) who had CMR imaging performed, with 21 (20%) of CMR imaging being performed prior to ICI initiation (median time from ICI to CMR was 204 days) and 83 (80%) of CMR imaging being performed after ICI initiation (median time from ICI to CMR was 511 days). CMR ordering indications are shown in **S4 Table**. Only 4 patients had CMR imaging both pre- and post-ICI; of these patients, 3 had non-diagnostic studies post-ICI, thus precluding evaluation of imaging changes pre- and post-ICI within the same patients. Quantitative hemodynamic measurements including LVEF, cardiac output, and cardiac index pre- and post-ICI are summarized in Table A of **S5 Table** and

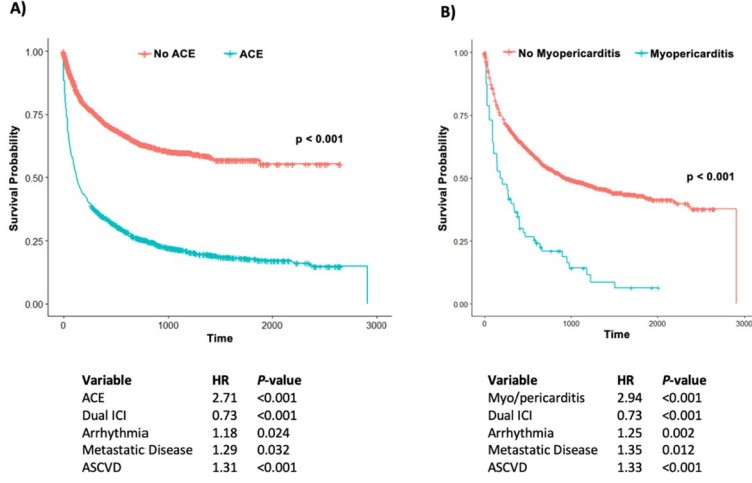

**Fig 2. Kaplan-Meier curves and time-dependent Cox regression for outcome of mortality.**

is notable for a significant decline in mean cardiac index post-ICI ($P$ = 0.024). Among patients who had CMR imaging prior to ICI initiation, 43% (n = 9/21) already had left ventricular dysfunction, 50% (n = 10/20) had right ventricular dysfunction, 32% (n = 6/19) had left ventricular LGE and 9% (n = 1/11) had abnormal T2 imaging. However, there were no significant differences in the prevalence of abnormal LVEF or RVEF, nor abnormal CMR tissue characteristics in comparing pre- and post-ICI scans (Table B of **S5 Table**).

Out of the 104 patients who had CMR performed, 14 (14%) had ICI myocarditis. Among patients who developed myocarditis, there was a significantly higher burden of any left ventricular LGE (61.5% vs 24.7%, $P$ = 0.018), non-ischemic left ventricular LGE (45.5% vs 7.7%, $P$ = 0.004), and right ventricular insertion LGE (38.5% vs 5.2%, $P$ = 0.003) compared to those who did not develop myocarditis (Table C of **S5 Table**). There was also a trend towards an increased burden of abnormal T2 in patients with myocarditis (33.3% vs 8.3%), although this did not meet statistical significance ($P$ = 0.063).

**Central illustration.** Risk Factors Increasing Adverse Cardiovascular Events and Cardiac Imaging findings in Patients Pre- and Post-ICI.

## Discussion

In this large single-center retrospective cohort study, we found an increased risk of ACE and myo/pericarditis with dual ICI therapy, which is consistent with what has been previously reported in the literature. A multicenter registry of 35 confirmed myocarditis cases in 8 medical centers in North America between 2013 and 2017 showed that myocarditis cases were more likely to have received combination ICI therapy at any stage in treatment compared to controls [22]. Similarly, analysis of a retrospective safety database conferred a 4.7-fold increased risk of developing myocarditis with combination of nivolumab and ipilimumab compared with treatment with nivolumab alone [9,13].

Patients in our cohort who developed ACE or myo/pericarditis had shorter survival. This was found to be independent of their comorbidities, cancer type, and age. Interestingly, we found that dual ICI therapy was protective against death—this may likely be in part due to improved cancer survival as prior studies have shown [47,48], despite the increased risk of ACE and associated mortality. Myo/pericarditis affected approximately 1% of this cohort, suggesting that the majority of patients likely still derive benefit from ICI therapy. Thus, shedding light upon additional risk factors for ICI myocarditis is crucial for risk stratification of higher risk patients [49].

It remains unclear if ICI associated cardiotoxicity is more common in specific types of cancers. In our cohort, patients with lung and CNS malignancies were at a higher risk of developing ACE. The observation of increased ACE with lung cancer may be in part related to close cardiopulmonary interactions; lung cancers can be implicated in the pathogenesis of pulmonary hypertension, pericardial effusions, and pleural effusions, which can in turn lead to heart failure-like symptoms. It is also possible that some patients with lung cancer received radiation therapy, which may have driven part of this observed effect. Neurologic complications from CNS malignancies may have promoted a proarrhythmic state, which could in turn contribute to higher rates of ACE. Although our study attempts to adjust for confounding factors as much as possible through multivariable analyses and deliberate covariate selection, it is unclear whether other unidentified shared risk factors may be playing a role in both cancer and progression of underlying cardiovascular disease. Further studies are needed to determine causality of these risk factors with the development of ACE on ICI therapy.

Although the role of CTLA-4 and PD-1 in regulating the inflammatory response underlying atherosclerosis is established [50], the effects of ICIs on atherosclerosis in cancer patients are

not completely understood [51]. For example, two retrospective studies on patients with non-small cell lung cancer (NSCLC) did not show a significant increase in ACE with ICI therapy [32,52]. Similarly, our propensity score matched analysis did not suggest a higher risk of ACE with ICI therapy in our cohort. In contrast, in a pooled analysis of 59 oncological trials comprising 21,664 patients, there was a trend towards increased coronary ischemia over 6 months follow up in patients who received ICIs compared to traditional cytotoxic chemotherapies [53]. Furthermore, in a single center retrospective study, authors reported a significant increase in the risk of MI, coronary revascularization, and ischemic CVA associated with the use of an ICI in a control-matched retrospective cohort study of 2842 patients. They also conducted a case-crossover analysis showing a significant increase in ischemic cardiovascular events within two years of initiation of ICIs [34]. In our propensity score matched analysis, it is possible that similar comorbidities and data supporting that TKIs may also be associated with increased ACE could explain the lack of difference between these two groups, who may both be at higher risk of developing ACE [54].

The 2022 European Society of Cardiology cardio-oncology guidelines recommend a baseline echo for patients starting ICI as a class I level recommendation for high-risk patients (patients starting dual ICI therapy, prior ICI related non-cardiovascular events, prior cardiovascular disease) and class IIb for those who are low risk [55]. Our study includes one of the largest published cohorts of patients on ICI therapy with cardiac imaging data that offers evidence to support this recommendation, given the high burden of imaging abnormalities observed in patients prior to ICI initiation. While it may be logistically challenging to obtain baseline CMR on every patient prior to ICI initiation, screening troponin, B-type natriuretic peptides, and echo may help delineate individuals who would derive the most benefit from having baseline CMR imaging. Establishing baseline cardiac imaging helps to identify individuals with pre-existing imaging abnormalities. This is not only important in informing risk of potential cardiotoxicity with ICI initiation, but is also of particular importance especially when a patient presents for evaluation of suspected ICI-related cardiotoxicity, as the clinician is afforded important comparative information, which can impact recommendations on whether to continue ICI therapy.

## Study strengths

Our study included a large cohort of over 2000 patients, providing robust statistical power and confidence in our findings. Moreover, we adjusted for confounders as much as possible using deliberate variable selection and multivariable models. Our study provides greater insight into a broad spectrum of ACE beyond myocarditis, which has been the most studied of the cardiovascular irAEs. ACE other than myocarditis were common in our cohort and were associated with increased mortality, necessitating increased vigilance toward the detection of these adverse events. Importantly, this study also highlighted risk factors that may increase ACE with ICI use, which may help to guide preventative therapies for these patients. Additionally, we performed an imaging substudy using echo and CMR data and comparing pre- and post-ICI scans, which uncovered a high burden of abnormal cardiac imaging pre-ICI as well as a higher burden of abnormalities detected on post-ICI imaging.

## Study limitations

This is a retrospective cohort study at a single center using electronic health record data, which can contain biases, inaccuracies associated with electronic health records and could be less representative of the entire cancer population. The primary limitation of this study is related to its retrospective nature, as confounding factors cannot be completely adjusted for. The reliance

on ICD codes for identification of variables and outcomes may have been subject to coding inaccuracies. We addressed these limitations by performing a manual chart adjudication for a randomly chosen subset of patients, and this showed good correlation without significant differences between the electronic health record data acquisition and chart review. Another study limitation is the lack of cardiac imaging data for the majority of patients. Additionally, there were essentially no patients who had both pre- and post-ICI CMR imaging, precluding assessment of CMR imaging changes within the same patients. Lastly, it is possible that patients with prior cardiac imaging were already at higher cardiovascular risk, which could account for the high burden of observed imaging abnormalities in this cohort.

## Conclusion

We performed a comprehensive assessment of ACE associated with ICIs in a single center retrospective study and found that dual ICI therapy was significantly associated with increased risk of composite ACE and myo/pericarditis. Consistent with prior oncology trial data, dual ICI therapy was associated with overall longer survival, despite an increased risk of ACE. Those with prior cardiac comorbidities including arrhythmia exhibited higher rates of ACE, and those who developed ACE or myo/pericarditis had significantly reduced survival. Patients with lung and CNS malignancies were more likely to develop ACE. Cardiac imaging prior to ICI initiation detected a high burden of abnormalities, suggesting higher cardiovascular risk in this patient population. Patients receiving ICI therapy stand to benefit from optimization of cardiovascular risk factors in order to attenuate the risk of developing cardiovascular complications; moreover, baseline and surveillance cardiac imaging can be considered in this high-risk patient population to provide additional risk stratification.

As a future direction, we plan to expand our cohort with more current data with the objective of constructing a robust model to estimate cardiovascular risk with ICI use. This model would incorporate patient characteristics, cardiovascular conditions, medical comorbidities, cancer information, as well as laboratory and cardiac imaging data, if available. Our goal is to eventually implement a risk score for widespread clinical use, which can provide patients who are being considered for ICI therapy with a tangible estimation of the potential cardiovascular risks, tailored to their unique risk factor profile.

## Supporting information

**S1 Fig. Propensity score matching for outcome of adverse cardiovascular events.**
(DOCX)

**S2 Fig. Kaplan Meier curves and median times to adverse cardiovascular events (ACE) and individual components of ACE.**
(DOCX)

**S3 Fig. Cox regression model including single immune checkpoint inhibitor agents for adverse cardiovascular events (ACE) and individual components of ACE.**
(DOCX)

**S4 Fig. Prevalence of abnormal echocardiograms pre- and post-immune checkpoint inhibitor.**
(DOCX)

**S1 Table. Comparison of demographic characteristics and comorbidities between entire cohort from electronic health record data acquisition and adjudicated subset.**
(DOCX)

**S2 Table. Presence of adverse cardiovascular events by cancer type.**
(DOCX)

**S3 Table. Demographics and comorbidities of propensity matched cohort.**
(DOCX)

**S4 Table. Ordering indications for cardiac magnetic resonance imaging pre- and post-immune checkpoint inhibitor.**
(DOCX)

**S5 Table. Cardiac magnetic resonance imaging features.**
(DOCX)

**S1 Data. Minimal dataset.**
(CSV)

**S1 File.**
(DOCX)

## Acknowledgments

We thank Krishna Daggula and Richard Hintz from the Yale JDAT team for their assistance with data acquisition.

## Author Contributions

**Conceptualization:** Jennifer M. Kwan, Miles Shen, Lauren A. Baldassarre.

**Data curation:** Jennifer M. Kwan, Miles Shen, Narjes Akhlaghi, Ruben Mora, Matthew Jiang, Michael Mankbadi, Peter Wang, Saif Zaman, Seohyuk Lee, Yunju Im, Yi-Hwa Liu, Shuangge S. Ma, Weiwei Tao, Wei Wei.

**Formal analysis:** Jennifer M. Kwan, Miles Shen, Narjes Akhlaghi, Ruben Mora, James L. Cross, Matthew Jiang, Yunju Im, Yi-Hwa Liu, Shuangge S. Ma, Weiwei Tao, Wei Wei, Lauren A. Baldassarre.

**Funding acquisition:** Jennifer M. Kwan, Lauren A. Baldassarre.

**Investigation:** Jennifer M. Kwan, Miles Shen, Narjes Akhlaghi, Ruben Mora, James L. Cross, Matthew Jiang, Michael Mankbadi, Peter Wang, Saif Zaman, Seohyuk Lee, Lauren A. Baldassarre.

**Methodology:** Jennifer M. Kwan, Miles Shen, Narjes Akhlaghi, James L. Cross, Matthew Jiang, Yunju Im, Yi-Hwa Liu, Shuangge S. Ma, Weiwei Tao, Wei Wei, Lauren A. Baldassarre.

**Project administration:** Jennifer M. Kwan, Lauren A. Baldassarre.

**Resources:** Jennifer M. Kwan, Miles Shen, Lauren A. Baldassarre.

**Software:** Yunju Im, Yi-Hwa Liu, Shuangge S. Ma, Weiwei Tao, Wei Wei.

**Supervision:** Jennifer M. Kwan, Lauren A. Baldassarre.

**Validation:** Jennifer M. Kwan, Miles Shen, Lauren A. Baldassarre.

**Visualization:** Jennifer M. Kwan, Miles Shen, Lauren A. Baldassarre.

**Writing – original draft:** Jennifer M. Kwan, Miles Shen, Narjes Akhlaghi, Matthew Jiang, Yunju Im.

**Writing – review & editing:** Jennifer M. Kwan, Miles Shen, Jiun-Ruey Hu, Ruben Mora, James L. Cross, Attila Feher, Lauren A. Baldassarre.

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
