## [Decision Letter · Decision Letter 0]

16 Sep 2024

PONE-D-24-31391Adverse Cardiovascular Events and Cardiac Imaging Findings in Patients on Immune Checkpoint InhibitorsPLOS ONE

Dear Dr. Baldassarre,

Thank you for submitting your manuscript to PLOS ONE. After careful consideration, we feel that it has merit but does not fully meet PLOS ONE’s publication criteria as it currently stands. Therefore, we invite you to submit a revised version of the manuscript that addresses the points raised during the review process.

We look forward to receiving your revised manuscript.

Kind regards,

Sai-Ching Jim Yeung, MD, PhD

Academic Editor

PLOS ONE

3. Please include your tables as part of your main manuscript and remove the individual files. Please note that supplementary tables (should remain/ be uploaded) as separate "supporting information" files.

4. We notice that your supplementary figures and tables are uploaded with the file type 'Figure'. Please amend the file type to 'Supporting Information'. Please ensure that each Supporting Information file has a legend listed in the manuscript after the references list.

Reviewers' comments:

Reviewer's Responses to Questions

**Comments to the Author**

1. Is the manuscript technically sound, and do the data support the conclusions?

Reviewer #1: Yes

Reviewer #2: Partly

2. Has the statistical analysis been performed appropriately and rigorously? 

Reviewer #1: Yes

Reviewer #2: I Don't Know

3. Have the authors made all data underlying the findings in their manuscript fully available?

Reviewer #1: Yes

Reviewer #2: Yes

4. Is the manuscript presented in an intelligible fashion and written in standard English?

Reviewer #1: Yes

Reviewer #2: Yes

5. Review Comments to the Author

Reviewer #1: This is a well written manuscript.

I wonder whether it is possible to develop a computer program based on your data and research which can calculate the pre-treatment risk and survival chance of an individual patient. This calculation could be similar to the heart score for chest pain with the calculation of a heart score which assigns a low risk, medium risk and high risk to the patient.

The risk factors could include dual versus single use of ICI, age, sex, arrhythmia , lung cancer and CNS cancer.

A score can certainly not predict the exact course of he ICI treatment but it could be helpful in the discussion with the patient searching for answers about treatment, probability of adverse effects of treatment and prognosis for treatment with ICI.

A newly developed score which can be computer generated could assist o find answers for the patient and the treating physicians when giving informed consent for ICI treatment. Additionally, table 2 A can be helpful to explain the risks of treatment and the fact that these adverse effects must be weighted towards the increased survival time with ICIs.

Maybe you can mention that based on your research you could try to develop such a computer program to calculate an ICI treatment score for risk stratification for adverse cardiac events within a certain period of time.

Supplemental figure 2 shows the Kaplan Meier curves and the times composite to ACE and individual composite to ACE.

Reviewer #2: SMOKING is a primary risk factor for ACE and represents a critical source of potential confounding in this study. Its absence from basic demographics of the cohort, as well as its absence in the propensity score matching, represents a significant flaw in this study. Any information on smoking status (previous, never, etc) is better than nothing to better understand how it fits in with pre/post ICI associative data.

Overall, this paper appears to emulate Drobni et al 2020, but does so with worse methodology (retrospective cohort rather than case-control, an even larger composite outcome, and with significant potential confounders that aren’t addressed with their propensity matching), and I think these issues should be addressed, if possible, with data re-analysis prior to publication (ex: data on previous cardiotoxic cancer therapies, and smoking status). I think this study adds a large body of data to support relevant associations such as dual ICI with increased risk ACE, mortality, and myo/pericarditis; there is thought-provoking data suggesting a not insignificant mortality benefit association from dual ICI (despite apparent associated risk of ACE/myo/pericarditis). It could still be worthy of publication but the data needs to be presented in a clearer way with less conflicting information. See below for line-by-line comments. Also, I am definitely not a statistician, so please excuse any errors I may have made and help me understand further if I have made mistakes. I wish you luck in improving your paper and hope these comments help get you closer to publication.

ABSTRACT:

Line 49: CNS malignancy is included as independently associated with ACE, however Supplemental Figure 3 seems to show “neuro” as crossing 1.0 in the top panel. Why do much of the data in that cox regression analysis in Supp. Fig. 3 conflict with the primary Figure 1 results? Is the primary difference that Supp. Fig. 3 included individual ICIs in the analysis? It would be helpful if the panels were formatted in the same orientation as Figure 1 so it makes it easier to compare differences. Overall, I’m not seeing the face validity of CNS malignancies being independently associated with ACE, and the presence of multiple conflicting points in the large amount of data graphics presented is a problem (ex: Figure 1 v. Supp. Fig. 3; “ACE” panel in Supp. Fig. 3 shows “neuro” crossing 1.0 suggesting statistical insignificance, yet Figure 1 shows “CNS malignancy” as statistically significantly associated with increased ACE; or “Arrhythmia” panel in Supp. Fig. 3. Shows “neuro” as actually protective from arrhythmias, yet Figure 1 shows “CNS malignancy” trending toward protective but ultimately crossing 1.0 – which is ironic because Line 318 in the Discussion section offers that “arrhythmias” may contribute to higher rates of ACE, despite the data not supporting that hypothesis at all, at least I’m not seeing support in Figure 1 or Supp. Fig. 3). Perhaps I’m misunderstanding something, however, so please help me understand. Also, if “neuro” in the Supp. Fig 3 is meant to be the same as “CNS malignancy” in Figure 1, then please edit the labels and reorient the panels so they appears as similar to Figure 1 as possible to make it easier to compare the two sets of Cox Regression Models.

Line 54: “19 of 90 who had CMR imaging…” Where did n=90 come from? Supplemental Table 4 shows ordering indications and suggests a total of 102 pre-ICI CMRs and 28 post-ICI CMRs… yet the supplemental Table 5 lists Pre-ICI total as n=90 and post-ICI CMRs as 14. Why is there a discrepancy between these numbers? What happened to the other 26 CMR scans?

Line 61: Again, “CNS malignancies” being included here needs to be supported by all of the data presented, and apparent conflict between Supp. Fig. 3 (“neuro”) and main Figure 1 (“CNS malignancies”) tables need to be explained.

MAIN PAPER:

Line 117: “…prior to and after ICI administration.” – this statement seems pretty misleading considering my comment below regarding Lines 265-270.

Line 124-125: Was the "Joint Data Analytics Team" (data abstractor) blinded to study outcome measures? Could a supplemental document noting specific ICD code/variable requests be made available?

Line 149-150: Why was “pericardial effusion” not a component of abnormal echo? To my knowledge, it’s part of the diagnostic criterion for pericarditis, which is one of the primary outcomes being examined in both Figure 1 and Table 1. I imagine there are a great deal of clinically insignificant pericardial effusions, however if they’re universally present (or absent) in cases of confirmed ICI-related myo/pericarditis then I think it’s work reporting. Why was “valvular disease” not a component of abnormal echo? It’s one of the primary outcome elements (“valvular disease [moderate or severe]”).

Line 171-172: I think a little more detail regarding coding techniques/keywords/instructions involved in using NLP in this way would be appreciated.

Line 192-193: In my anecdotal experience, I see ipilimumab being utilized more often as part of a dual ICI strategy (ex: ipi/nivo) than as a mono-ICI. That’s in contrast to frequently seeing nivolumab monotherapy (for whatever reason). Does the data in Table 2 isolate instances of ipilimumab as only monotherapy, or could recorded instances of its use for dual ICI be potentially driving its appearance as an outlier in that list?

Line 199: Why are the “n” values so large and varied in this list? It would appear obviously much larger than total cohort of 2165. Please help me understand where these total values are coming from.

Line 210-212: How is prior ASCVD associated with a lower risk of ASCVD following ICI therapy? This seems completely counterintuitive and potentially a data error of some kind. This requires further explanation.

Line 227: see my first comment above (Line 49) that has concerns regarding Supplemental Figure 3.

Line 228-230: “after adjusting for covariates in a multivariable Cox regression model, it was not independently associated with…” need to add a bolded reference to Supplemental Figure 3 at the end of this sentence.

Line 237-238: “However, dual ICI therapy was found to decrease mortality by ~25%.” I see in line 299-305 the authors suggest that low incidence of myocarditis suggests “the majority of patients likely still [derive] mortality benefit from ICI therapy.” – if this is correct, then perhaps this findings is worthy of inclusion in the abstract for the paper.

Line 249: Supplemental Table 3 – please edit the table so the variable presented match Table 1 (ex: either switch Table 1 to be alphabetical, or re-order Supplemental Table 3 to more closely match the order of variables presented in Table 1).

Line 258: “Supplemental Figure 4” – please add “n” information to the infographic to clearly see how many echo exams were performed.

Line 265: Why does Supplemental Table 4 indicate a total of 130 CMR exams (102 pre-ICI, 28 post-ICI) that were ordered? Were 26 CMR studies ordered but never completed? Perhaps add bolded reference to Supplemental Table 5 at the end of this sentence on line 268.

Line 265: “(5%)” perhaps add clarification of some kind such as “104 of the entire cohort of 2165 (5%) had CMR performed…”

Line 265: “21 (20%) of CMR imaging being performed prior to ICI initiation…” – Supp. Table 5 suggests 90/104 (87%) were performed prior to ICI initiation. Why the discrepancy?

Line 267: “…83 (80%) of CMR images being performed after ICI initiation…” – Supp. Table 5 suggests 14/104 (13%) were performed post-ICI initiation. Why the discrepancy?

Line 268-270: If only 4 patients had CMR imaging both pre/post ICI and only 1 had diagnostic value… perhaps the entire “substudy” portion involving CMR imaging should be re-framed to focus solely on the increased LGE post-ICI (one of the only parts of Supp. Table 5 to have proper significance). I’m not familiar with “RV Insertion LGE” but that appeared to have p<0.05 in Supp. Table 5.

Line 273: “Supplemental Table 5B” – Abnormal T1 findings are part of the “Lake Louise Criteria” referenced in Line 113, how come that variable is absent in this table?

Line 277: Supp. Table 5 indicates T2 findings did not achieve statistical significance (P=0.063), perhaps add a qualifying statement to this sentence explicitly stating lack of p-value significance.

Line 287: Central Illustration – “Outcomes” in the bottom right portion of the figure – is there a way to keep the units equivalent when comparing the relative pros/cons of dual ICI? It’s tough to compare “~2x” or “~3x” with “25%.”

Line 312-315: this is a huge limitation, and calls to question the validity of the propensity score matching in this dataset.

Line 350-351: why wasn’t troponin or BNP included in this dataset?

Line 364: the absence of “smoking” and “previous cardiotoxic treatments” suggests it wasn’t adjusted as much as possible. Surely some information on these confounders are extractable from the EHR.

Line 365: “…with echo and CMR obtained prior to and after ICI initiation” – again, somewhat misleading verbiage, please find a way to word this to clearly state cardiac imaging was obtained when clinically indicated, and all pre-ICI scans were compared with post-ICI scans, though almost none of them were in the same patient.

Line 372: consider adding that “ACE” is a composite outcome with composite elements (ex: ASCVD) above and beyond what is typically considered “MACE.” This will probably make this data very difficult to

Line 377: is 10% individual chart review good enough? Is there a way to demonstrate that ~200 charts of 2000 is sufficiently powered to make the conclusion that there are no significant differences between the EHR data pull and chart review?

Line 392: if “metastatic disease” is highlighted in the main body of the conclusion, I would recommend highlighting that in the abstract rather than “CNS malignancy.”

6. PLOS authors have the option to publish the peer review history of their article (what does this mean?). If published, this will include your full peer review and any attached files.

Reviewer #1: **Yes: **Tareg Bey, MD

Reviewer #2: **Yes: **Jonathan Rowland

---

## [Author Response · Author response to Decision Letter 0]

30 Oct 2024

See attached "Response to Reviewers" document

---

## [Editor Report · Decision Letter 1]

13 Nov 2024

Adverse Cardiovascular Events and Cardiac Imaging Findings in Patients on Immune Checkpoint Inhibitors

PONE-D-24-31391R1

Dear Dr. Baldassarre,

We’re pleased to inform you that your manuscript has been judged scientifically suitable for publication and will be formally accepted for publication once it meets all outstanding technical requirements.

Kind regards,

Sai-Ching Jim Yeung, MD, PhD

Academic Editor

PLOS ONE
---

## [Editor Report · Acceptance letter]

18 Nov 2024

PONE-D-24-31391R1 

PLOS ONE

Dear Dr. Baldassarre, 

I'm pleased to inform you that your manuscript has been deemed suitable for publication in PLOS ONE. Congratulations! Your manuscript is now being handed over to our production team.

Kind regards, 

on behalf of

Dr. Sai-Ching Jim Yeung 

Academic Editor

PLOS ONE